# Cooperative Games Based on Coalition Functions in Biform Games

**Chenwei Liu [1], Shuwen Xiang [1,2,\*] , Yanlong Yang [1] and Enquan Luo [3]**

1. School of Mathematics and Statistics, Guizhou University, Guiyang 550025, China
2. College of Mathematical and Information Science, Guiyang University, Guiyang 550005, China
3. School of Management, Guizhou University, Guiyang 550025, China
* Correspondence: shwxiang@vip.163.com or swxiang@gzu.edu.cn

**Abstract:** In this paper, we try to study a class of biform games with the coalition function from the cooperation of players. For this purpose, we interpret the biform games as cooperative games by defining a characteristic function of minimax representation based on the coalition function and giving the core and Shapley value as cooperative solutions. The relations between the coalition function and the characteristic function are investigated in terms of additivity and convexity, and the properties associated with the characteristic function, such as individual rationalities and cores, are compared with the corresponding results. The relations among the solutions of the normal-form game, biform game, and cooperative game are discussed with several examples.

**Keywords:** cooperative games; Shapley value; core; characteristic functions; additivity

## 1. Introduction

The concept of transferable utility (TU)-game is one of the important game theories. In the aspect that TU-games depend on the strategic choice of players, Ui [1] introduced a model of TU-games with action choices and focused on reducing a TU-game with action choices to a strategic game where the payoff of each strategy profile is determined by the Shapley value of the corresponding TU-game. In the same line, Brandenburger and Stuart [2] proposed a class of biform games where the value of each coalition depends on the strategies of all players. Fiestras–Janeiro et al. [3] established a TU-game with strategies where the value of a coalition is affected by the strategies of players outside that coalition.

$(A_1, \cdots, A_n; V; \alpha_1, \cdots, \alpha_n)$ is the biform game model proposed by Brandenburger and Stuart [2], where $A_i$ is a finite strategy set of player $i$, $i \in N = \{1, \cdots, n\}$; $V$ is a map from $A = \prod_{i=1}^{n} A_i$ to the set of maps from $2^N$ (a set of all coalitions) to the reals, for each strategy profile $c \in A$, $V(c)(S)$ is the value created by the coalition $S$, with $V(c)(\varnothing) = 0$ (in this paper, $V(c)(S)$ is written as $V_S(c)$, and the map $V$ is referred to as the coalition function); for each $i \in N$, the number $\alpha_i (0 \leq \alpha_i \leq 1)$ is player $i$'s confidence index. Brandenburger and Stuart gave a detailed explanation for the confidence index.

The biform game is a hybrid noncooperative–cooperative game model with two stages. The first stage is noncooperative and is designed to describe the strategic moves of the players. However, the consequences of these moves are not the utilities (payoffs) of all players, but the utilities of all coalitions, thus each profile of strategic choices at the first stage leads to a second stage. The second is cooperative and is designed to analyze how to allocate utility to each player on each strategy profile. In general, the core [4] or the Shapley value [5] is used for this analysis. The cooperative second stage yields an induced noncooperative game that ultimately presents the Nash equilibrium through the strategic

moves of players in the first stage. Therefore, the overall tendency of the biform game is noncooperative. Stuart [6] pointed out that the biform game is a noncooperative game.

There is a significant difference between the biform game [2] and the normal-form game [7], that is, on each strategy profile, the normal-form game generates the utilities (payoffs) for all players, whereas the biform game generates the utilities for all coalitions. This difference is well-identified by biform models and examples in the literature of Grossman and Hart [8], Hart and Moore [9], Brandenburger and Stuart [2], Summerfield and Dror [10], Gonzalez et al. [11], and Grove et al. [12].

This correspondence between strategy profiles and utilities of players or coalitions implies that the coalition function $V$ has multiple additive properties. If coalition function $V$ is additive on each strategy, then the game is in normal form. It has the usual noncooperative Nash equilibrium [13] and cooperative solutions (such as the core [4], $\alpha$- and $\beta$-core [14,15], and Shapley value [5]). To consider the cooperative solutions of a normal-form game, there are three main ways to establish the characteristic functions: the minimax representation [16], the defensive-equilibrium representation [17], and the rational-threats representation [18]. The characteristic function form of a normal-form game is usually defined as $(N, v)$ [17], where $v$ is the minimax representation.

In the TU-game model with strategies proposed by Fiestras–Janeiro et al. [3], similar to the characteristic functions above, they defined the "maxmin procedure" and gave the corresponding core as the cooperative solution. This analysis implies that players can cooperate and coordinate their strategic power to allocate the utility of the grand coalition.

For a game based on the coalition function $V$, if the cooperative game theory is applied, then the characteristic function should be given for the whole game pattern rather than each strategy profile. This paper is just based on this idea to give cooperative solutions for a class of biform games. For this purpose, we define a characteristic function of minimax representation $v$ based on the coalition function $V$, interpret biform games as cooperative games, and give the core and Shapley value as cooperative solutions. Based on $V$ and $v$, we establish a cooperative game model in the characteristic function form $(N, V, v)$. The characteristic function of minimax representation we defined is different from the "maxmin procedure" given by Fiestras–Janeiro et al. [3], and thus the corresponding cores are also different.

The main contributions of this work are summarized as follows: First, this paper presents the concept of cooperative game solutions for a class of biform games with the coalition function $V$. The key to this concept is to establish a characteristic function $v$ based on the coalition function $V$. This concept is different from the existing results of the biform games. The technical route of the biform game is: first, on each strategy profile, the utility functions of all players are obtained from the core or Shapley value of the cooperative game determined by the coalition function $V$; second, the Nash equilibrium of the noncooperative game is taken as a solution of the biform game, where the noncooperative game is determined by the utility functions on the set of strategy profiles. The technical route of this paper is to directly give the characteristic function $v$, build a cooperative game model $(N, V, v)$, and then get the cooperative solutions, where the utility $v(S)$ of a coalition is obtained by taking the maximum and minimum in the strategy changes of the insiders and outsiders of this coalition, respectively, through the coalition function $V$. Second, we show the superadditivity and convexity of $v$ by the corresponding properties of $V$ and extend the properties of the core and Shapley value of game $(N, v)$ to the corresponding solutions of game $(N, V, v)$. Third, the choices of game models and their solutions are shown based on the additivity of $V$. By observing the solutions of the biform game and cooperative game $(N, V, v)$, we find the advantages of cooperative game solutions, namely that they are always collectively rational and easier and faster to compute.

The remainder of this paper is organized as follows. Section 2 defines the characteristic function $v$, establishes a cooperative game model $(N, V, v)$, and examines the relation between "minmax" and "maxmin". Section 3 studies the relations between the characteristic function $v$ and coalition function $V$ and defines the core and Shapley value

of the cooperative game $(N, V, v)$. The solutions of the cooperative games and biform games are compared in Section 4. The applicability of Shapley value and the two stages of the cooperative game $(N, V, v)$ are discussed in Section 5. Some final summaries are in Section 6.

## 2. The Model

Let $N = \{1, \cdots, n\}$ be the set of players and $2^N$ be the set of subsets (i.e., coalitions) of $N$. For each nonempty coalition $S \in 2^N$, the lowercase letter $s$ denotes the number of players in coalition $S$. $X_i = \{x_{i1}, \cdots, x_{im_i}\}$ is the finite strategy set of player $i \in N$ and $X = \prod_{i=1}^{n} X_i$ is the set of strategy profiles. For each nonempty coalition $S \in 2^N$, let $X_S = \{x_S : x_S = (x_{1t_1}, \cdots, x_{it_i}, \cdots, x_{st_s}), x_{it_i} \in X_i, i \in S\}$ be the set of strategies of coalition $S$, and $x_S$ is a strategy of coalition $S$ (see Section 9.2 of Myerson [17]). For each $i \in N$ and nonempty $S \subseteq N$, let $-i = N \backslash \{i\}$ and $-S = N \backslash S$, then $x = (x_S, x_{-S}) \in (X_S, X_{-S}) = X$.

The coalition function $V$ is a map from $X$ to the set of maps from $2^N$ to the reals. For each strategy profile $x \in X$, the utility of coalition $S \in 2^N$ is given by the map $V(x) : 2^N \to \mathbb{R}$, that is, $V_S(x)$ is the utility created by coalition $S$ on $x$, in particular, $V_{\varnothing}(x) = 0$ for any $x \in X$.

The definition of the coalition function is extracted from the biform games, existing research results, and application examples, such as the studies of Hart and Moore [9], Brandenburger and Stuart [2], Gonzalez et al. [11], and Grove et al. [12]. The coalition function reflects the nature of the correspondence between the strategy profiles and the coalition utilities or player utilities, that is, either the utilities of the players or the utilities of the coalitions are generated on each strategy profile.

Let $V$ be a coalition function. For any coalitions $S_1, S_2 \in 2^N$ with $S_1 \cap S_2 = \varnothing$.

(1) We say that $V$ is *superadditive* on $x \in X$ if $V_{S_1 \cup S_2}(x) \geq V_{S_1}(x) + V_{S_2}(x)$ and $V$ is *superadditive* on $X$ if it is superadditive for any $x \in X$;

(2) We say that $V$ is *subadditive* on $x \in X$ if $V_{S_1 \cup S_2}(x) \leq V_{S_1}(x) + V_{S_2}(x)$ and $V$ is *subadditive* on $X$ if it is subadditive for any $x \in X$; and

(3) We say that $V$ is *additive* on $x \in X$ if $V_{S_1 \cup S_2}(x) = V_{S_1}(x) + V_{S_2}(x)$ and $V$ is *additive* on $X$ if it is additive for any $x \in X$.

The feature of the coalition function $V$, generating each coalition $S$ utility $V_S(x)$ on each strategy profile $x$, provides convenience for players to form coalitions. Each coalition $S$ has the rationality to choose a strategy to maximize its utility, given a strategy of the complementary coalition $-S$. We refer to Von Neumann and Morgenstern [16] and define the characteristic function of *minimax representation* $v : 2^N \to \mathbb{R}$ as

$$v(S) = \min_{x_{-S} \in X_{-S}} \max_{x_S \in X_S} V_S(x_S, x_{-S}), \forall S \in 2^N.$$

In particular, $v(N) = \max_{x \in X} V_N(x)$ is the maximum utility created by the grand coalition $N$.

This definition asserts that $v(S)$ is the maximum utility that the members of coalition $S$ can guarantee themselves against the best offensive threat by the complementary coalition $-S$. Please refer to Myerson's [17] formula (9.1) for the characteristic function definition of the strategic game with TU (i.e., the normal-form game with TU).

Let $G^N$ be the set of the characteristic functions $v$ with the usual operations of addition and scalar multiplication of functions in a $(2^N - 1)$-dimensional linear space.

**Definition 1.** *An $n-$person cooperative game in the characteristic function form is defined as*

$$(N, V, v),$$

*where*

*(1) V is the coalition function;*

*(2)    v is the characteristic function of minimax representation.*

The characteristic function of minimax representation is distinguished from the "maxmin procedure" of Fiestras–Janeiro et al. [3]. Let the coalition $S$'s utility determined by the "maxmin procedure" be

$$\psi(S) = \max_{x_S \in X_S} \min_{x_{-S} \in X_{-S}} V_S(x_S, x_{-S})$$

for all $S \in 2^N$.

For each $S \in 2^N$, it is clear that

$$\min_{x_{-S} \in X_{-S}} \max_{x_S \in X_S} V_S(x_S, x_{-S}) \geq \max_{x_S \in X_S} \min_{x_{-S} \in X_{-S}} V_S(x_S, x_{-S}),$$

that is, $v(S) \geq \psi(S)$. In Table 1, it is easy to get $v(\{3\}) = 0 > -3 = \psi(\{3\})$ and $v(\{1,2\}) = 4 > -1 = \psi(\{1,2\})$. Therefore, we know that the equation in $v(S) \geq \psi(S)$ is not necessarily true.

If coalition function $V$ is superadditive on $X$, the Nash equilibrium can be given by the biform game [2,19]; the core and the Shapley value (see next section) can be given by cooperative game $(N, V, v)$.

If $V$ is additive on $X$, the Nash equilibria of the biform game and the normal-form game are identical; the cooperative game $(N, V, v)$ becomes the cooperative game $(N, v)$, the core and the Shapley value of $(N, v)$ are the special form of the solutions of $(N, V, v)$.

If $V$ is subadditive on $X$, there exists no normal-form game. Since the core on each strategy profile is an empty set, the biform game [2] is not applicable in this case. The solution to the second stage of the biform game can be the Shapley value, thereby generating an induced noncooperative game to obtain the Nash equilibrium. Alternatively, we may consider the cooperative game $(N, V, v)$ to deal with this situation (see Example 3).

As can be seen, based on the additivity of the coalition function $V$, the games in normal form, biform games, and cooperative games $(N, V, v)$ and $(N, v)$ can be linked to form a systematic framework for strategic form games.

The example below comes from Example 2 of Summerfield and Dror [10].

**Table 1.** Utilities of coalitions for the game (1).

|  | (0,0,0) | (0,1,0) | (1,0,0) | (1,1,0) | (0,0,1) | (0,1,1) | (1,0,1) | (1,1,1) |
|---|---|---|---|---|---|---|---|---|
| $\{1\}$ | 0 | 0 | 2 | 2 | 0 | 0 | 2 | 2 |
| $\{2\}$ | 0 | −1 | 0 | −1 | 0 | −1 | 0 | −1 |
| $\{3\}$ | 0 | 0 | 0 | 0 | −3 | −3 | −3 | −3 |
| $\{1,2\}$ | 0 | −1 | 2 | 4 | 0 | −1 | 2 | 4 |
| $\{1,3\}$ | 0 | 0 | 2 | 2 | −3 | −3 | 3 | 3 |
| $\{2,3\}$ | 0 | −1 | 0 | −1 | −3 | 2 | −3 | 2 |
| $\{1,2,3\}$ | 0 | −1 | 2 | 4 | −3 | 2 | 3 | 4.4 |

**Example 1.** *Consider the case of a business owner who approaches three advertising firms, each with its own cable channel, to produce and broadcast an advertising program. Each firm has to decide independently whether or not to accept the job offer. The business owner is willing to pay $17 to each firm for his/her job. In the game (1) (see Table 1), the three advertising firms are players 1, 2, and 3, each one with a strategy set $\{0, 1\}$, indicating that one is turning down the job or, respectively, accepting the job offer. The three strategies of each strategy profile in the game (1) are owned by players 1, 2, and 3 in turn. Let $C(S)$ denote the total cost of creating advertisements art works for firms in coalition $S$. The following is the complete cost schedule for advertisement(s) collaboration costs, assuming all firms take their assigned jobs.*

$$C(\{1\}) = 15, C(\{2\}) = 18, C(\{3\}) = 20, C(\{1,2\}) = 30,$$
$$C(\{1,3\}) = 31, C(\{2,3\}) = 32, C(\{1,2,3\}) = 46.6.$$

*For each strategy profile c of this game, the utility $V_S(c)$ of coalition S is equal to the income paid by the business to S minus the cost of S for creating advertisement art work. For instance,*

$$V_{\{1\}}((0,1,1)) = 0 \times 17 - 0 = 0,$$
$$V_{\{2\}}((0,1,1)) = 1 \times 17 - 18 = -1,$$
$$V_{\{2,3\}}((0,1,1)) = 2 \times 17 - 32 = 2.$$

*Table 1 shows the utilities of coalitions on strategy profiles, it is obvious that the utility of each player cannot be generated directly on each strategy profile, but rather the utilities of all coalitions are generated, each player cannot directly observe his/her utility on strategy profiles except $(0,0,0)$.*

*By Table 1, V is additive on $(0,0,0)$, $(0,1,0)$, $(1,0,0)$ and $(0,0,1)$. It is superadditive on other strategy profiles.*

*If the players need to determine their respective utility on each strategy profile, then they can play cooperatively and use the core or Shapley value as the solution, this is the second stage of the biform game proposed by Brandenburger and Stuart [2], that is, the cooperative games.*

*We should note that players have the willingness to form coalitions to improve the utilities of themselves and the coalitions, and coincidentally, this behavior is facilitated by the fact that all coalitions' utilities are generated on each strategy profile. In this way, each coalition chooses the strategy that benefits its utility, given that its complementary coalition chose a strategy. In Table 1, when coalition $\{3\}$ chooses strategy 0, the relevant strategy profiles are $(0,0,0)$, $(0,1,0)$, $(1,0,0)$, and $(1,1,0)$, thus, the maximum utility of coalition $\{1,2\}$ is $max\{0,-1,2,4\} = 4$; it is easy to find that when coalition $\{3\}$ chooses strategy 1, the maximum utility of coalition $\{1,2\}$ is also 4. Therefore,*

$$v(\{1,2\}) = min\{4,4\} = 4.$$

*Similarly, we obtain*

$$v(\{1\}) = 2, v(\{2\}) = v(\{3\}) = 0, v(\{1,3\}) = 3,$$
$$v(\{2,3\}) = 2, v(\{1,2,3\}) = 4.4.$$

*Based on the characteristic function values, each player's utility can be determined by the cooperative solution core and Shapley value (see next section).*

The above example provides a practical application context for the coalition function $V$, the characteristic function $v$, and the cooperative game model $(N, V, v)$.

### 3. Main Results

*3.1. The Relations of V and v*

Let $v$ be the characteristic function of minimax representation of an $n-$person cooperative game $(N, V, v)$. For any $S_1, S_2 \in 2^N, S_1 \cap S_2 = \varnothing$, $v$ is said to be *superadditive* if $v(S_1 \cup S_2) \geq v(S_1) + v(S_2)$; $v$ is said to be *subadditive* if $v(S_1 \cup S_2) \leq v(S_1) + v(S_2)$; and $v$ is said to be *additive* if $v(S_1 \cup S_2) = v(S_1) + v(S_2)$.

The superadditivity and convexity of characteristic function $v$ are important contents of the cooperative game theory. The following results show these properties based on coalition function $V$.

**Theorem 1.** *If V is superadditive on X, then v is also superadditive.*

**Proof.** For any $S_1, S_2 \in 2^N, S_1 \cap S_2 = \varnothing$, since $S_2 \cup -(S_1 \cup S_2) = -S_1$ and $S_1 \cup -(S_1 \cup S_2) = -S_2$, we have

$$v(S_1) = \min_{x_{-S_1} \in X_{-S_1}} \max_{x_{S_1} \in X_{S_1}} V_{S_1}(x_{S_1}, x_{-S_1})$$

$$= \min_{x_{S_2 \cup -(S_1 \cup S_2)} \in X_{S_2 \cup -(S_1 \cup S_2)}} \max_{x_{S_1} \in X_{S_1}} V_{S_1}(x_{S_1}, x_{-S_1}),$$

$$v(S_2) = \min_{x_{-S_2} \in X_{-S_2}} \max_{x_{S_2} \in X_{-S_2}} V_{S_2}(x_{S_2}, x_{-S_2})$$

$$= \min_{x_{S_1 \cup -(S_1 \cup S_2)} \in X_{S_1 \cup -(S_1 \cup S_2)}} \max_{x_{S_2} \in X_{S_2}} V_{S_2}(x_{S_2}, x_{-S_2}).$$

In $v(S_1)$ and $v(S_2)$, let the "maximum strategies" taken by coalition $S_1, S_2$ be $\bar{x}_{S_1}, \bar{x}_{S_2}$, respectively, then for any fixed $x_{-(S_1 \cup S_2)} \in X_{-(S_1 \cup S_2)}$, we have

$$\min_{x_{S_2 \cup -(S_1 \cup S_2)} \in X_{S_2 \cup -(S_1 \cup S_2)}} \max_{x_{S_1} \in X_{S_1}} V_{S_1}(x_{S_1}, x_{-S_1}) \leq V_{S_1}(\bar{x}_{S_1}, \bar{x}_{S_2}, x_{-(S_1 \cup S_2)}),$$

$$\min_{x_{S_1 \cup -(S_1 \cup S_2)} \in X_{S_1 \cup -(S_1 \cup S_2)}} \max_{x_{S_2} \in X_{S_2}} V_{S_2}(x_{S_2}, x_{-S_2}) \leq V_{S_2}(\bar{x}_{S_1}, \bar{x}_{S_2}, x_{-(S_1 \cup S_2)}).$$

By making $V$ superadditive on $X$, we get

$$v(S_1) + v(S_2) \leq V_{S_1}(\bar{x}_{S_1}, \bar{x}_{S_2}, x_{-(S_1 \cup S_2)}) + V_{S_2}(\bar{x}_{S_1}, \bar{x}_{S_2}, x_{-(S_1 \cup S_2)})$$

$$\leq V_{S_1 \cup S_2}(\bar{x}_{S_1}, \bar{x}_{S_2}, x_{-(S_1 \cup S_2)}).$$

In $v(S_1 \cup S_2) = \min_{x_{-(S_1 \cup S_2)} \in X_{-(S_1 \cup S_2)}} \max_{x_{S_1 \cup S_2} \in X_{S_1 \cup S_2}} V_{S_1 \cup S_2}(x_{S_1 \cup S_2}, x_{-(S_1 \cup S_2)})$, let the "minimum strategy" determined by $-(S_1 \cup S_2)$ be $\check{x}_{-(S_1 \cup S_2)}$. Therefore,

$$v(S_1) + v(S_2) \leq V_{S_1 \cup S_2}(\bar{x}_{S_1}, \bar{x}_{S_2}, \check{x}_{-(S_1 \cup S_2)})$$

$$\leq \max_{x_{S_1 \cup S_2} \in X_{S_1 \cup S_2}} V_{S_1 \cup S_2}(x_{S_1 \cup S_2}, \check{x}_{-(S_1 \cup S_2)})$$

$$= v(S_1 \cup S_2).$$

This completes the proof. □

**Corollary 1.** *If $V$ is additive on $X$, then $v$ is also superadditive.*

**Proof.** By the proof of Theorem 1 and $V$ is additive on $X$, we have

$$v(S_1) + v(S_2) \leq V_{S_1}(\bar{x}_{S_1}, \bar{x}_{S_2}, \check{x}_{-(S_1 \cup S_2)}) + V_{S_2}(\bar{x}_{S_1}, \bar{x}_{S_2}, \check{x}_{-(S_1 \cup S_2)})$$

$$= V_{S_1 \cup S_2}(\bar{x}_{S_1}, \bar{x}_{S_2}, \check{x}_{-(S_1 \cup S_2)})$$

$$\leq \max_{x_{S_1 \cup S_2} \in X_{S_1 \cup S_2}} V_{S_1 \cup S_2}(x_{S_1 \cup S_2}, \check{x}_{-(S_1 \cup S_2)})$$

$$= v(S_1 \cup S_2).$$

This completes the proof. □

In Example 1, $v$ is superadditive by the superadditivity of $V$ on $X$.

**Remark 1.** *If $V$ is subadditive on $X$, then for $v$ there are various possibilities of additivity (see Example 3).*

If $V$ is subadditive on $X$, then the superadditive of $v$ is given by the following theorem.

**Theorem 2.** *If V is subadditive on X, and*

$$V_{S_1}(\bar{x}_{S_1}, \bar{x}_{S_2}, \check{x}_{-(S_1 \cup S_2)}) + V_{S_2}(\bar{x}_{S_2}, \bar{x}_{S_1}, \check{x}_{-(S_1 \cup S_2)})$$
$$\in [V_{S_1 \cup S_2}(\bar{x}_{S_1}, \bar{x}_{S_2}, \check{x}_{-(S_1 \cup S_2)}), V_{S_1 \cup S_2}(\bar{x}_{S_1 \cup S_2}, \check{x}_{-(S_1 \cup S_2)})],$$

*then v is superadditive. Where $\bar{x}_{S_1}, \bar{x}_{S_2}$, and $\bar{x}_{S_1 \cup S_2}$ are the "maximum strategies" selected by coalitions $S_1, S_2$ and $S_1 \cup S_2$ in $v(S_1), v(S_2)$ and $v(S_1 \cup S_2)$, respectively, and $\check{x}_{-(S_1 \cup S_2)}$ is the "minimum strategy" selected by coalition $-(S_1 \cup S_2)$ in $v(S_1 \cup S_2)$.*

**Proof.** For any $S_1, S_2 \in 2^N, S_1 \cap S_2 = \varnothing$, from the known conditions, we get

$$v(S_1) + v(S_2) \leq V_{S_1}(\bar{x}_{S_1}, \bar{x}_{S_2}, \check{x}_{-(S_1 \cup S_2)}) + V_{S_2}(\bar{x}_{S_2}, \bar{x}_{S_1}, \check{x}_{-(S_1 \cup S_2)})$$
$$\in [V_{S_1 \cup S_2}(\bar{x}_{S_1}, \bar{x}_{S_2}, \check{x}_{-(S_1 \cup S_2)}), V_{S_1 \cup S_2}(\bar{x}_{S_1 \cup S_2}, \check{x}_{-(S_1 \cup S_2)})],$$

therefore,

$$v(S_1) + v(S_2) \leq v(S_1 \cup S_2).$$

This completes the proof. □

In Table on page 11, $V$ is subadditive on $X$. It is easy to get $\bar{x}_{\{1\}} = a_1$ and $\bar{x}_{\{2\}} = a_2$, then $V_{\{1\}}(a_1, a_2) = 2, V_{\{2\}}(a_1, a_2) = 1, V_{\{1,2\}}(a_1, a_2) = 2, v_{\{1,2\}} = 4$. Therefore, $V$ satisfies Theorem 2 and $v$ is superadditive, where $-\{1, 2\}$ is an empty set.

For any $S_1, S_2 \in 2^N$, an $n-$person cooperative game $(N, V, v)$ is said to be *convex* [20,21] if

$$v(S_1 \cup S_2) \geq v(S_1) + v(S_2) - v(S_1 \cap S_2);$$

$V$ is said to be *convex* on $x \in X$ if

$$V_{S_1 \cup S_2}(x) \geq V_{S_1}(x) + V_{S_2}(x) - V_{S_1 \cap S_2}(x);$$

$V$ is said to be *convex* on $X$ if $V$ is convex for any $x \in X$.

**Theorem 3.** *If V is convex on X and*

$$V_{S_1 \cap S_2}(\bar{x}_{S_1}, \bar{x}_{S_2}, \check{x}_{-(S_1 \cup S_2)}) \leq v(S_1 \cap S_2),$$

*then an $n-$person cooperative game $(N, V, v)$ is also convex. Where $\bar{x}_{S_1}$ and $\bar{x}_{S_2}$ are the "maximum strategies" selected by coalitions $S_1$ and $S_2$ in $v(S_1)$ and $v(S_2)$, respectively, and $\check{x}_{-(S_1 \cup S_2)}$ is the "minimum strategy" selected by coalition $-(S_1 \cup S_2)$ in $v(S_1 \cup S_2)$.*

**Proof.** For any $S_1, S_2 \in 2^N$, by the definition of $v$, we have

$$v(S_1) \leq V_{S_1}(\bar{x}_{S_1}, \bar{x}_{S_2}, \check{x}_{-(S_1 \cup S_2)}),$$
$$v(S_2) \leq V_{S_2}(\bar{x}_{S_1}, \bar{x}_{S_2}, \check{x}_{-(S_1 \cup S_2)}).$$

Since $V$ is convex on $X$, we get

$$V_{S_1}(\bar{x}_{S_1}, \bar{x}_{S_2}, \check{x}_{-(S_1 \cup S_2)}) + V_{S_2}(\bar{x}_{S_1}, \bar{x}_{S_2}, \check{x}_{-(S_1 \cup S_2)})$$
$$\leq V_{S_1 \cup S_2}(\bar{x}_{S_1}, \bar{x}_{S_2}, \check{x}_{-(S_1 \cup S_2)}) + V_{S_1 \cap S_2}(\bar{x}_{S_1}, \bar{x}_{S_2}, \check{x}_{-(S_1 \cup S_2)}).$$

Thus, let $\bar{x}_{S_1 \cup S_2}$ be the "maximum strategies" taken by coalition $S_1 \cup S_2$ in $v(S_1 \cup S_2)$, we have

$$v(S_1) + v(S_2) \leq V_{S_1 \cup S_2}(\bar{x}_{S_1}, \bar{x}_{S_2}, \check{x}_{-(S_1 \cup S_2)}) + V_{S_1 \cap S_2}(\bar{x}_{S_1}, \bar{x}_{S_2}, \check{x}_{-(S_1 \cup S_2)})$$
$$\leq V_{S_1 \cup S_2}(\bar{x}_{S_1 \cup S_2}, \check{x}_{-(S_1 \cup S_2)}) + V_{S_1 \cap S_2}(\bar{x}_{S_1}, \bar{x}_{S_2}, \check{x}_{-(S_1 \cup S_2)})$$
$$\leq v(S_1 \cup S_2) + V_{S_1 \cap S_2}(\bar{x}_{S_1}, \bar{x}_{S_2}, \check{x}_{-(S_1 \cup S_2)}),$$

and since

$$V_{S_1 \cap S_2}(\bar{x}_{S_1}, \bar{x}_{S_2}, \check{x}_{-(S_1 \cup S_2)}) \leq v(S_1 \cap S_2),$$

Therefore,

$$v(S_1) + v(S_2) - v(S_1 \cap S_2) \leq v(S_1 \cup S_2).$$

This completes the proof. □

It is easy to verify that if $V$ is convex on $X$, there must be $V$ that is superadditive on $X$. The converse is not true. By Theorem 1, we have the following corollary.

**Corollary 2.** *If $V$ is convex on $X$, then $v$ is superadditive.*

**Remark 2.** *Theorems 1, 2 and Corollary 1 show that the characteristic function $v$ defined by "minmax" can maintain the additivity of the coalition function $V$ under certain conditions. The consistency of the additivity of $v$ and $V$ ensures that the cooperative solution has good properties, and also shows that the cooperation is reasonable.*

*3.2. The Core and Shapley Value*

Given an $n-$person cooperative game $(N, V, v)$, the utility allocation $u = (u_1, \cdots, u_n)$ $\in \mathbb{R}^n$ is said to be *individual rational* if $u_i \geq v(\{i\})$ for all $i \in N$, it is said to be *collective rational* if $\sum_{i \in N} u_i = v(N)$, and it is said to be *coalition rational* if $\sum_{i \in S} u_i \geq v_S(x)$ for any $S \in 2^N$.

For an $n-$person cooperative game $(N, V, v)$, its *core* [4] is the set of utility allocations that satisfy collective rationality and coalition rationality, that is,

$$C(v) = \{u = (u_1, \cdots, u_n) \in \mathbb{R}^n : \sum_{i \in N} u_i = v(N); \sum_{i \in S} u_i \geq v(S), \forall S \in 2^N\}.$$

Its *Shapley value* $\varphi(v) \in \mathbb{R}^n$ [5] is given by

$$\varphi_i(v) = \sum_{S \subseteq -i} \frac{s!(n-s-1)!}{n!}[v(S \cup \{i\}) - v(S)]$$

for all $i \in N$.

Let $\pi(N)$ be the set of all permutations $\sigma : N \to N$, for each permutation $\sigma \in \pi(N)$, denote $\sigma = \sigma(1), \cdots, \sigma(n)$. Then $\varphi(v) = (\varphi_{\sigma(1)}(v), \cdots, \varphi_{\sigma(n)}(v))$.

For any nonempty coalition $S \in 2^N$, denote by $1^S \in \mathbb{R}^n$ the characteristic vector of $S$, where its $i-$th coordinate is

$$(1^S)_i = \begin{cases} 1, & \text{if } i \in S, \\ 0, & \text{otherwise.} \end{cases}$$

A map $\lambda : 2^N \backslash \{\varnothing\} \to \mathbb{R}_+$ is called a *balanced map* if $\sum_{S \in 2^N \backslash \{\varnothing\}} \lambda(S) 1^S = 1^N$. A cooperative game $(N, V, v)$ is said to be *balanced* if $\sum_{S \in 2^N \backslash \{\varnothing\}} \lambda(S) v(S) \leq v(N)$ for each balanced map $\lambda$.

For each $v \in G^N$, the Shapley value $\varphi(v)$ satisfies

(1) *efficiency* if $\sum_{i=1}^{n} \varphi_i(v) = v(N)$;

(2) *dummy player property* if $\varphi_i(v) = v(\{i\})$ for any dummy players $i \in N$, player $i$ is a dummy player means that $v(S \cup \{i\}) = v(S) + v(\{i\})$ for all $S \in 2^{-i}$;

(3) *anonymity property* if $\varphi_{\sigma(i)}(\sigma v) = \varphi_i(v)$ for any $\sigma \in \pi(N)$, any $v \in G^N$, and all $i \in N$, where $\sigma v(\{\sigma(i), i \in S\}) = v(S)$ for all $S \in 2^N \backslash \{\varnothing\}$;

(4) *additivity* if $\varphi_i(v + w) = \varphi_i(v) + \varphi_i(w)$ for any $v, w \in G^N$.

Now we give the following results:

(1) An $n-$person cooperative game $(N, V, v)$ has a nonempty core $C(v)$ if and only if it is balanced. It can be proved by referring to Proposition 262.1 in Chapter 13 of Osborne and Rubinstein [22].

(2) For each $v \in G^N$, the Shapley value $\varphi(v)$ is the unique solution that satisfies efficiency, dummy player property, anonymity property, and additivity. It can be proved by referring to Shapley [5] and Proposition 3.5 in Chapter 3 of Branzei et al. [23].

(3) If cooperative game $(N, V, v)$ is convex, then $C(v)$ is nonempty, by referencing Proposition 28.1 in Chapter 28 of Narahari [24].

(4) We also obtain that if cooperative game $(N, V, v)$ is convex, then $\varphi(v) \in C(v)$, by $C(v)$ is nonempty, according to Exercises 260.4 and 295.5 of Osborne and Rubinstein [22].

After generating the characteristic function $v$ based on the coalition function $V$, the proofs of the above results are only related to $v$, not $V$.

**Remark 3.** *The proofs of the above results are easy. However, it is worth noting that these results are based on the coalition function V, and this means that these results take the corresponding results in the usual strategic form game with TU (i.e., the normal-form game with TU) as the special form, because V has various possibilities of additivity on X and the usual strategic form game with TU only corresponds to V with additivity on X.*

**Theorem 4.** *If V is convex on X and*

$$V_{S_1 \cap S_2}(\bar{x}_{S_1}, \bar{x}_{S_2}, \check{x}_{-(S_1 \cup S_2)}) \leq v(S_1 \cap S_2),$$

*Then:*

*(1) $(N, V, v)$ is balanced;*

*(2) $\varphi(v) \in C(v)$.*

*where $\bar{x}_{S_1}$ and $\bar{x}_{S_2}$ are the "maximum strategies" selected by coalitions $S_1$ and $S_2$ in $v(S_1)$ and $v(S_2)$, respectively, and $\check{x}_{-(S_1 \cup S_2)}$ is the "minimum strategy" selected by coalition $-(S_1 \cup S_2)$ in $v(S_1 \cup S_2)$.*

**Proof.** From Theorem 3, $(N, V, v)$ is a convex game. Thus, $C(v)$ is nonempty. Therefore: (1) $(N, V, v)$ is balanced; (2) $\varphi(v) \in C(v)$. This completes the proof. □

**Property 1.** *If V is superadditive on X, then the Shapley value $\varphi(v)$ is individual rational.*

**Proof.** By Theorem 1, $v$ is superadditive. Then, for any $S \in 2^N$,

$$v(S \cup \{i\}) - v(S) \geq v(\{i\}),$$

and since $\varphi_i(v) = \sum_{S \subseteq -i} \dfrac{s!(n-s-1)!}{n!} = 1$, we have

$$\varphi_i(v) = \sum_{S \subseteq -i} \frac{s!(n-s-1)!}{n!}[v(S \cup \{i\}) - v(S)] \geq v(\{i\})$$

for all $i \in N$. Therefore, $\varphi(v)$ is individual rational. This completes the proof. □

After Definition 1, the meaning of $\psi(S)$ is described. Here, we write the core corresponding to $\psi(S)$ as $C(\psi)$ to show the following remark. Please refer to Fiestras–Janeiro et al. [3] for relevant results of this core.

**Remark 4.** *(1) $v(S) \geq \psi(S)$ for all single-player coalition $S \in 2^N$ means that the individual rationality requirement corresponding to v is not lower than that corresponding to $\psi$. This shows that grand coalition N is likely to has less surplus to allocate, then, it is easier to obtain an allocation*

*with small allocation difference in the utility allocations corresponding to v than in the utility allocations corresponding to ψ;*

　　*(2) By the definitions of $C(v)$ and $C(\psi)$, it is easy to verify that $C(v) \subseteq C(\psi)$.*

## 4. Comparison of the Solutions

**Example 2.** *Consider a 3−person game (2) [2], each player has two strategies, labelled "No" and "Yes", the three strategies in each strategy profile are owned by players 1, 2, and 3, respectively, Table 2 shows the coalition utilities of the game. The utilities of all single-player coalitions are taken as zero, so their utilities were not entered in Table 2.*

　　*By the utilities of all single-player coalitions, we get $v(\{1\}) = v(\{2\}) = v(\{3\}) = 0$. When player 3 chooses "No", the maximum utility of coalition $\{1,2\}$ is 6, when player 3 chooses "Yes", the maximum utility of coalition $\{1,2\}$ is also 6, thus, $v(\{1,2\}) = min\{6,6\} = 6$. Similarly, we have $v(\{1,3\}) = v(\{2,3\}) = 6$, $v(\{1,2,3\}) = 9$. Therefore,*

$$C(v) = \{(u_1, u_2, u_3) \in \mathbb{R}^3 : 0 \le u_1 \le 3, 0 \le u_2 \le 3, 0 \le u_3 \le 3; \sum_{i=1}^{3} h_i = 9\},$$

$$\varphi_1(v) = 2/6 \times 0 + 1/6 \times 6 + 1/6 \times 6 + 2/6 \times (9 - 6) = 3,$$
$$\varphi_2(v) = \varphi_3(v) = 3.$$

**Table 2.** Utilities of coalitions for the 3−person game (2).

|  | (No, No, No) | (No, Yes, No) | (Yes, No, No) | (Yes, Yes, No) |
|---|---|---|---|---|
| $\{1,2\}$ | 4 | 3 | 3 | 6 |
| $\{1,3\}$ | 4 | 4 | 3 | 3 |
| $\{2,3\}$ | 4 | 3 | 4 | 3 |
| $\{1,2,3\}$ | 6 | 5 | 5 | 6 |
|  | (No, No, Yes) | (No, Yes, Yes) | (Yes, No, Yes) | (Yes, Yes, Yes) |
| $\{1,2\}$ | 4 | 3 | 3 | 6 |
| $\{1,3\}$ | 3 | 3 | 6 | 6 |
| $\{2,3\}$ | 3 | 6 | 3 | 6 |
| $\{1,2,3\}$ | 5 | 6 | 6 | 9 |

　　Brandenburger and Stuart used the biform game analysis to obtain two Nash equilibria of the induced noncooperative game (3) (this game is induced by the cores, see Table 3): (No, No, No) and (Yes, Yes, Yes), the corresponding utility allocations are $(2, 2, 2)$ and $(3, 3, 3)$. In the 3−person game (2), the most likely core allocation is $(3, 3, 3)$, and the Shapley value is also $(3, 3, 3)$. Therefore, the Nash equilibrium (No, No, No) is inefficient (the total value is 6), while (Yes, Yes, Yes) is efficient (the total value is 9), and the cooperative solution $(3, 3, 3)$ is collective rational.

**Table 3.** A 3−person induced noncooperative game (3).

| | | **Player 2 × Player 3** | | | | | |
|---|---|---|---|---|---|---|---|
| | | No, No | Yes, No | | | No, Yes | Yes, Yes |
| Player 1 | No | 2, 2, 2 | 2, 1, 2 | | | 2, 2, 1 | 0, 3, 3 |
| | Yes | 1, 2, 2 | 3, 3, 0 | | | 3, 0, 3 | 3, 3, 3 |

　　In Example 1, we have

$$C(v) = \{(u_1, u_2, u_3) \in \mathbb{R}^3 : 2 \le u_1 \le 2.4, 0 \le u_2 \le 1.4, 0 \le u_3 \le 0.4; \sum_{i=1}^{3} u_i = 4.4\}$$

$$\varphi_1(v) = 2/6 \times 2 + 1/6 \times 4 + 1/6 \times 3 + 2/6 \times (4.4 - 2) = 79/30,$$
$$\varphi_2(v) = 34/30, \varphi_3(v) = 19/30.$$

Summerfield and Dror [10] developed an approach that treats a biform game as a two-stage stochastic programming problem with recourse, this approach deals with either empty or non-empty "Core games". In the game of Example 1, Summerfield and Dror gave two Nash equilibria $(1, 0, 0)$ and $(1, 1, 1)$ depending on the range of confidence indices, the corresponding utility allocations are $(2, 0, 0)$ and $(6\alpha + 2(1 - \alpha), 5\beta - (1 - \beta), 3\gamma - 3(1 - r))$, where $\alpha, \beta, \gamma$ are respectively the confidence indices of Firms 1, 2, and 3, and $0 \leq \alpha \leq 1, 1/3 < \beta \leq 1, 3/4 < \gamma \leq 1$. This approach means that the utility of each player is determined by the subgames and the confidence indices, this leads to multiple Nash equilibria. This paper adopts the method of the cooperative game $(N, V, v)$, the characteristic function reflects the global rational behavior of players. Moreover, it makes the solution process simpler and faster, because the players are not required to enter the subgame and determine the confidence indices. The Shapley value in this example gives a fair result, which is an allocation worthy of serious consideration by the players. As with the usual core of the cooperative game $(N, v)$, the core given in this example can also capture the idea of freeform competition among the players.

To sum up, in the games with additivity of coalition function: (1) The solutions of the cooperative game $(N, V, v)$ may achieve the same result as the efficient Nash equilibrium (the utility allocation on it is collective rational) of a biform game. (2) The Shapley value of $(N, V, v)$ provides a fair and unique allocation vector. The core of $(N, V, v)$ satisfies individual rationality and collective rationality, and it also captures the idea of free-form competition among the players. In effect, the players can compete in the core by combining the fair outcome of the Shapley value. (3) If the core is an empty set on a strategy profile, then the cooperative stage of the biform game cannot be used the core. In this case, the Shapley value can be applied on any strategy profile to generate an induced noncooperative game to obtain the Nash equilibrium. In addition, to get a game solution, the Shapley value of the cooperative game $(N, V, v)$ can be used (see Example 3).

**Example 3.** *Consider a* $2-$*person game (4),* $A_i = \{a_i, b_i\}$ *is the strategy set of player i* $(i = 1, 2)$*. Table 4 shows the utilities of coalitions on strategy profiles.*

*If a biform game is considered for the solution, then in the second stage, the core on each strategy profile is an empty set, thus, the Nash equilibrium cannot be obtained. If the Shapley value is considered on each strategy profile, then the induced noncooperative game (5) is shown in Table 5. This noncooperative game has a Nash equilibrium* $(b_1, b_2)$*, the corresponding utility allocation is* $(5/2, 3/2)$*.*

*By the method of the cooperative game* $(N, V, v)$*, we have*

$$v(\{1\}) = 2, v(\{2\}) = 1, v(\{1, 2\}) = 4,$$
$$C(v) = \{(u_1, u_2) \in \mathbb{R}^2 : 2 \leq u_1 \leq 3, 1 \leq u_2 \leq 2, u_1 + u_2 = 4\},$$
$$\varphi_1(v) = 1/2 \times 2 + 1/2 \times 3 = 5/2, \varphi_2(v) = 3/2.$$

*Therefore, players can obtain the utility allocation* $(5/2, 3/2)$*.*

*By Table 4, v is supperadditive. If coalition* $\{1, 2\}$*'s utility is* $29/10$ *on* $(b_1, b_2)$ *and other coalition utilities remain unchanged, then v is subadditive; if* $29/10$ *is replaced by 3, v becomes additive. Therefore, when V is subadditive, v is superadditive only if V satisfies the certain condition, as shown in Theorem 2.*

**Table 4.** Utilities of coalitions for $2-$person game (4).

|            | $(a_1, a_2)$ | $(b_1, a_2)$ | $(a_1, b_2)$ | $(b_1, b_2)$ |
|------------|--------------|--------------|--------------|--------------|
| $\{1\}$    | 2            | 1            | 1            | 3            |
| $\{2\}$    | 1            | 1            | 1            | 2            |
| $\{1, 2\}$ | 2            | 3/2          | 3/2          | 4            |

**Table 5.** A 2−person induced noncooperative game (5).

| | | Player 2 | |
|---|---|---|---|
| | | $a_2$ | $b_2$ |
| Player 1 | $a_1$ | $3/2, 1/2$ | $3/4, 3/4$ |
| | $b_1$ | $3/4, 3/4$ | $5/2, 3/2$ |

## 5. Discussions

*5.1. CIS Value of the Cooperative Game $(N, V, v)$*

If only the utilities of single-player coalitions and the grand coalition are generated on each strategy profile, then the Shapley value $\varphi(v)$ cannot allocate utility for each player. In this case, we consider the CIS value (Driessen and Funaki, [25]). The CIS value is a centre of the imputation set value that first allocates the utility of working individually to each player, and then allocates the remaining utility of the grand coalition equally. In Definition 3 of Herrero et al. [26], this value is also called the rights egalitarian solution.

For a cooperative game $(N, V, v)$, its *CIS value $CIS(v) \in \mathbb{R}^n$* is given by

$$CIS_i(v) = v(\{i\}) + \frac{1}{n}[v(N) - \sum_{j \in N} v(\{j\})]$$

for all $i \in N$.

**Example 4.** *There are three miners 1, 2, and 3 in a mining pool, each miner has the strategies 0 and 1, 0 is a malicious strategy and 1 is an honest strategy. The three strategies in each strategy profile are owned by miners 1, 2, and 3 in turn. Table 6 shows the utilities of coalitions on strategy profiles.*

**Table 6.** Utilities of coalitions for 3−person game (6).

| | (0, 0, 0) | (0, 1, 0) | (1, 0, 0) | (1, 1, 0) | (0, 0, 1) | (0, 1, 1) | (1, 0, 1) | (1, 1, 1) |
|---|---|---|---|---|---|---|---|---|
| $\{1\}$ | 0 | 0 | 0.0375 | 0.0375 | 0 | 0 | 0.0375 | 0.0375 |
| $\{2\}$ | 0 | 0.1375 | 0.0019 | 0.1375 | 0.0019 | 0.1375 | 0.0019 | 0.1375 |
| $\{3\}$ | 0 | 0 | 0 | 0 | 0.01 | 0.01 | 0.01 | 0.01 |
| $\{1, 2, 3\}$ | 0 | 0.4225 | 0.3315 | 0.559 | 0.0145 | 0.5135 | 0.4225 | 0.65 |

*According to the biform game analysis, Du et al. [27] applied SCIS value to generate an induced noncooperative games (see their Table 5) to obtain a Nash equilibrium $(1, 1, 1)$, corresponding to the utility allocation $(0.2355, 0.2322, 0.2035)$.*

*Since*

$$v(\{1\}) = 0.0375, v(\{2\}) = 0.1375, v(\{3\}) = 0.01, v(\{1, 2, 3\}) = 0.65,$$

*the Shapley value is inapplicable for calculating players' utilities, while CIS value is applicable. Thus, we have*

$$CIS_1(v) = 0.0375 + 1/3[0.65 - (0.0375 + 0.1375 + 0.01)] = 0.1925,$$
$$CIS_2(v) = 0.2925, CIS_3(v) = 0.165.$$

*$v(\{1\}) = 0.0375$ and $v(\{2\}) = 0.1375$ indicate that player 2 has better utility than player 1 when working alone. Moreover, from the utilities of all coalitions in Table 6, it is easy to find that player 2 contributes more to the coalitions than player 1. Thus, the utility allocation $(0.1925, 0.2925, 0.165)$ is more reasonable than utility allocation $(0.2355, 0.2322, 0.2035)$.*

*5.2. Two Stages of the Cooperative Game $(N, V, v)$*

The two stages of the cooperative game $(N, V, v)$ are explained as follows.

In the first stage, the coalition function values are generated on each strategy profile. Note that some players have the cooperation willingness to form a coalition. The coalition chooses the strategy to improve its utility, given a strategy of its complementary coalition. Therefore, this stage is the cooperative game stage. The strategy profile, determined by each pair of complementary coalitions, affects the utilities of players in the second stage. The second stage is also the cooperative game stage, where the core or Shapley value gives each player's utility based on all coalitions' utilities in the first stage. The first stage finally obtains the characteristic function values, then, the second stage gives the final utility for each player by the core or Shapley value.

In the first stage of $(N, V, v)$, if there exists $(x_S, x_{-S}) \in X$ and a coalition $S \in 2^N$ such that $v(S) \neq V_S(x_S, x_{-S})$, that is, in all $(2^n - 1)$ nonempty coalition function values, as long as one is not the characteristic function value, then in the second stage, the *core* is denoted as

$$C(V) = \{u = (u_1, \cdots, u_n) \in \mathbb{R}^n : \sum_{i \in N} u_i = \max_{x \in X} V_N(x) = v(N);$$

$$\sum_{i \in S} u_i \geq V_S(x_S, x_{-S}), \forall S \in 2^N\},$$

and the *Shapley value* $\varphi(V) \in \mathbb{R}^n$ is given by

$$\varphi_i(V) = \sum_{S \subseteq -i} \frac{s!(n-s-1)!}{n!} [V_{S \cup \{i\}}(x_{S \cup \{i\}}, x_{-(S \cup \{i\})}) - V_S(x_S, x_{-S})]$$

for all $i \in N$.

In the first stage of the game in Example 1, if coalition $\{1\}$ chooses strategy 0, no matter which strategy the coalition $\{2, 3\}$ chooses, the utility of coalition $\{1\}$ is 0 ($0 \neq v(\{1\})$). The utilities of other coalitions still take the characteristic function values, then, the second stage has

$$C(V) = \{(u_1, u_2, u_3) \in \mathbb{R}^3 : 0 \leq u_1 \leq 2.4, 0 \leq u_2 \leq 1.4, 0 \leq u_3 \leq 0.4; \sum_{i=1}^{3} u_i = 4.4\},$$
$$\varphi_1(V) = 2/6 \times 0 + 1/6 \times 4 + 1/6 \times 3 + 2/6 \times 2.4 = 59/30,$$
$$\varphi_2(V) = 44/30, \varphi_3(V) = 29/30.$$

Compare $C(V)$, $\varphi(V)$ with their corresponding $C(v)$, $\varphi(v)$ in Section 4, this causes player 1 to be less competitive in $C(V)$ because the individual rationality value is reduced, it also reduces the utility of player 1 in the Shapley value $\varphi(V)$. If player 1 chooses strategy 1, then the strategy would lead to a greater utility $\varphi_1(v)$. In this case, all coalition function values determined by $v$ are in a stable state, that is, no coalition can change its strategy to increase its utility and change the utility of any other coalition. Therefore, the utility allocation given by the Shapley value $\varphi(v)$ in the second stage of $(N, V, v)$ is the final solution vector.

In the first stage of the game in Example 2, if coalition $\{1, 2\}$ takes strategy (No, Yes), then the utility of coalition $\{1, 2\}$ is 3 regardless of which strategy is taken by player 3. Similarly, if coalition $\{1, 3\}$ takes strategy (No, No), coalition $\{1, 3\}$'s utility is 4, and if coalition $\{2, 3\}$ takes strategy (No, No), $\{2, 3\}$'s utility is also 4. Other coalitions' utilities still take the characteristic function values. Then

$$C(V) = \{(u_1, u_2, u_3) \in \mathbb{R}^3 : 0 \leq u_1 \leq 5, 0 \leq u_2 \leq 5, 0 \leq u_3 \leq 6; \sum_{i=1}^{3} u_i = 9\},$$
$$\varphi_1(V) = 2/6 \times 0 + 1/6 \times 3 + 1/6 \times 4 + 2/6 \times 5 = 17/6,$$
$$\varphi_2(V) = 17/6, \varphi_3(V) = 20/6.$$

Consider a $C(V)$, there may be a situation where the game cannot be coordinated, in this situation, the utility created by the grand coalition cannot meet the allocation requirement of the players, because each player may require the utility obtained to be greater than 3. While the core under the characteristic function is reasonable, in the $C(v)$,

each player can obtain utility 3. As can be seen from the symmetry of the players' strategy contributions, the utility allocation in $\varphi(V)$ is unfair. Thus, all players will adjust their strategies, and the final result is also a stable state as mentioned above, that is, through the strategy choices of each pair of complementary coalitions, each coalition obtains the value of the characteristic function.

## 6. Conclusions

This paper proposed the notion of cooperative solutions for a class of biform games containing the coalition function $V$. The main method was to build the characteristic function of minimax representation $v$ based on the coalition function $V$, define the cooperative game $(N, V, v)$, and give the solutions core and Shapley.

The definition of $v$ refers to the characteristic function of the usual strategic game with TU. The superadditivity of $v$ can be obtained from various additive properties of $V$ under certain conditions. This guarantees that cooperative solutions can have good properties, and also shows the rationality of cooperation. The core corresponding to $v$ refines the core corresponding to the characteristic function defined by the "maxmin procedure". $v$ raises the individual rationality requirement of $\psi$ and reduces the surplus that grand coalition $N$ needs to allocate, thus reducing the allocation difference.

The choices of the solutions of biform games, cooperative games, and noncooperative games were presented in terms of the additivity of the coalition function $V$. Through several examples, the solutions of cooperative games and biform games were compared. It was found that the solution of the cooperative game can be the utility allocation on the efficient Nash equilibrium of the biform game.

**Author Contributions:** Conceptualization, C.L. and S.X.; methodology, C.L. and S.X.; software, C.L. and S.X.; validation, C.L., S.X., Y.Y. and E.L.; writing—original draft preparation, C.L.; writing review and editing, C.L. and S.X.; visualization, C.L., S.X., Y.Y. and E.L.; supervision, C.L., S.X., Y.Y. and E.L.; project administration, S.X. and Y.Y.; funding acquisition, S.X. and Y.Y. All authors have read and agreed to the published version of the manuscript.

**Funding:** This research was supported by National Natural Science Foundation of China (Grant No. [71961003], [12061020]), Qian Ke He LH (Grant No. [2017] 7223), and Talent Introduction Foundation of Guizhou University (Grant No. [(2019) 49]).

**Institutional Review Board Statement:** Not applicable.

**Informed Consent Statement:** Not applicable.

**Data Availability Statement:** The data sets used and/or analyzed during the current study are available from the corresponding author on reasonable request.

**Acknowledgments:** The authors are grateful to the referees for their careful reading of the manuscript and valuable comments. The authors thank the help from the editor too.

**Conflicts of Interest:** The authors declare no conflicts of interest.

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
