# Peer review of "Cooperative Games Based on Coalition Functions in Biform Games"

_axioms, doi:10.3390/axioms12030296_

Round 1

Reviewer 1 Report

I have to say that I had a hard time reading the paper. I think the subject is interesting but it is not well described from the beginning. Here are some suggestions that may help the authors to improve the paper: 

1. The abstract should clearly and concisely describe the content of the paper and the main contributions of the paper to the literature.

2. I do not see the need to present an example before describing the model. I think it would be more appropriate to use the example to illustrate the model and the main results. 

3.  Section 3 describes the model and shows some properties. I recommend explaining in a little more detail the definitions given as well as the properties obtained. 

4.  Section 4 seems to introduce some preliminary concepts. If so, it should go at the beginning. 

5. From reading the paper in its current version, it is not clear what are the results obtained. 

I encourage the authors to rewrite the paper completely in a way that motivates and presents their contributions to the biform games literature. 

Reviewer 2 Report

There are a lot of language and grammar issues which are distracting and confusing: line 54, should read "allocate utility to each player"; lines 59-60, unclear sentence; lines 136-138, unclear; line 194, unclear; line 203, write "... that V must be convex ..."; line 290, incomplete sentence; next line, messy verb phrase; and many more. 

Line 32: Notation V(c)(S) is strange notation. 

Line 59, line 87 and elsewhere: What is the "trend" of a game?

The role of the "confidence index" alpha never becomes clear. 

Line 295: Here Phi is used instead of alpha. Why? 

Apart from serious polishing this paper needs to do more to explain the notion and use of biform games in the introduction. An explanation like "The biformity of biform games is 'noncooperation' and 'cooperation' " does not help a lot.  

Lines 326-327: Notice that CIS (what does it stand for?) is discussed in the social choice literature under the name "rights egalitarian solution" (see Herrero et al. (1999) in Mathematical Social Sciences 37, pp. 59-77). 

Reviewer 3 Report

The paper is interesting, the results are clear and the proofs are correct. Proposition 2, as Corollario 1, is trivial and it must be change to Corollario. Theorem 1 and Theorem 2 are not interesting. As the author says they are trivial. Perhaps in the future the authors must look for conditions over V equivalents to be the core of v non empty, or an axiomatization of the Shapley value of v using axioms from V. But the paper introduces a nice method to analyze the biform problems and those theorems are not necessary.

Round 2

Reviewer 1 Report

I believe that the paper in its current state is not publishable in Axioms. 

Round 3

Reviewer 1 Report

The authors have made a great effort to satisfactorily address all the issues raised by the reviewers. I believe the paper is now publishable in its current form in Axioms.